# Adherence to Mediterranean Diet and Biomarkers of Redox Balance and Inflammation in Old Patients Hospitalized in Internal Medicine

**DOI:** 10.3390/nu16193359

**Published:** 2024-10-02

**Authors:** Francesco Bellanti, Aurelio Lo Buglio, Michał Dobrakowski, Aleksandra Kasperczyk, Sławomir Kasperczyk, Gaetano Serviddio, Gianluigi Vendemiale

**Affiliations:** 1Department of Medical and Surgical Sciences, University of Foggia, Viale Pinto 1, 71122 Foggia, Italy; francesco.bellanti@unifg.it (F.B.); aurelio.lobuglio@unifg.it (A.L.B.); gaetano.serviddio@unifg.it (G.S.); 2Department of Biochemistry, Faculty of Medical Sciences in Zabrze, Medical University of Silesia, 41-808 Katowice, Poland; michal.dobrakowski@poczta.fm (M.D.); olakasp@poczta.onet.pl (A.K.); skasperczyk@sum.edu.pl (S.K.)

**Keywords:** Mediterranean diet, inflammation, immune response, aging

## Abstract

Background/Objectives: We have previously described that low adherence to the Mediterranean diet (MD) in elderly patients admitted in internal medicine wards is linked to poorer clinical outcomes. This investigation was designed to explore whether adherence to the MD is related to circulating markers of redox balance and inflammation in this clinical scenario. Methods: A cross-sectional study was performed on 306 acute old patients hospitalized in internal medicine wards. Adherence to the MD was estimated by the Italian Mediterranean Index (IMI). The circulating markers of redox balance were assessed in serum and erythrocytes and correlated with inflammatory markers across different MD adherence groups. Results: Compared to the patients with high adherence, those with low adherence to the MD exhibited severely impaired redox balance, as evidenced by a higher GSSG/GSH ratio and increased serum hydroxynonenal/malondialdehyde–protein adducts. No modifications were described in the expression of antioxidant enzymes in peripheral blood mononuclear cells. Patients with low adherence to the MD exhibited a higher neutrophil-to-lymphocyte ratio and markers of systemic inflammation, as well as raised levels of interleukin-6 and tumor necrosis factor, compared to those with high MD adherence. A strong association was observed between the circulating markers of redox balance and inflammation/immune response, with the highest regression coefficients found in the low adherence group. Conclusions: Old patients admitted to internal medicine wards with low adherence to the MD display unfavorable profiles of the circulating markers of redox balance and inflammation. It is conceivable that such effects on redox balance can be linked to the high polyphenol content of MD. This study supports the rationale for intervention trials that attest to the effectiveness of MD as a nutritional strategy for disease prevention.

## 1. Introduction

The Mediterranean diet (MD) emphasizes the intake of “healthy” foods, such as fish, legumes, cereals, vegetables, fruits, and extra-virgin olive oil, alongside moderate amounts of red wine and restricted consumption of meat and meat products, high-fat milk, and dairy products [1,2]. This diet is notable to be associated with reduced mortality from cardiovascular diseases and its benefits for conditions including type 2 diabetes, obesity, neurodegenerative diseases, and particular cancer types [3,4,5,6,7]. The positive effects of the MD are thought to stem from its antioxidant, anti-inflammatory, blood pressure-lowering, cholesterol-lowering, and insulin-sensitizing properties [8,9,10,11].

Although traditionally prevalent in the Mediterranean region, adherence to the MD has significantly declined since the 1960s due to the westernization of dietary habits [12]. This decline is observed even in Mediterranean countries despite the well-documented health benefits and improved quality of life associated with the MD, especially for older adults [13].

Aging is often accompanied by a low-grade chronic inflammation, which is characterized by increased levels of acute phase proteins and pro-inflammatory cytokines, including interleukin 6 (IL-6) and tumor necrosis factor (TNF) [14,15,16,17]. This phenomenon, known as “inflammaging”, is linked to increased mortality and morbidity in the elderly. Inflammaging can also contribute to reduced appetite and food intake, a condition referred to as “anorexia of aging” [18,19]. 

Alteration of redox balance, known as oxidative stress, is recognized as a key contributor to the aging process. It is considered one of the hallmarks of aging and plays a key role in the pathological pathways that drive various age-related diseases [20]. Described as a disproportion between the formation of oxidative molecules and reducing compounds (antioxidants) in favor of the former, oxidative stress extends from physiological amounts, necessary for redox signaling, to a toxic degree that causes molecular or organelle impairment [21]. An excess of reactive species is associated with the process of inflammaging, contributing to the activation of cell death pathways that underlie several age-related diseases [22]. 

We have previously reported that low adherence to the MD in old patients hospitalized in internal medicine wards is linked to longer hospital stays and unfavorable circulating pro-inflammatory markers [23]. However, the relationship between the MD and redox balance needs to be defined in this particular context. This investigation was intended to define the association of adherence to the MD with the circulating markers of redox balance while also exploring possible associations with systemic inflammation.

## 2. Materials and Methods

### 2.1. Study Population and Design

This investigation was performed at the Department of Internal and Aging Medicine, “Policlinico Riuniti”, in Foggia, Italy. A total of 769 successive patients aged 65 years or older who were hospitalized from May 2022 to April 2023 were assessed for eligibility. The exclusion criteria included age <65 years, smoker status, alcohol consumption of >40 g/day, immune system disease, severe kidney or liver failure, dysphagia, active cancer, intake of drugs with potential impact on redox balance, severe cognitive impairment (defined as a Mini-Mental State Examination score of ≤9 points), and incapability or unwillingness to adhere to the study protocol or give written informed consent. Demographic information (age and gender), lifestyle aspects (smoking and alcohol consumption), chronic conditions (diabetes mellitus, hypertension, hyperlipidemia, chronic liver and/or kidney disease), and pharmacotherapy data were recorded.

The study received approval from the Institutional Ethics Committee of the Policlinico Riuniti in Foggia and was performed in conformity with the Declaration of Helsinki. Relevant data were collected anonymously in a dedicated database, and written informed consent was acquired from all participants. 

We used the Italian Mediterranean Index (IMI) to assess MD adherence. The IMI score is calculated based on the consumption of the following 11 items: high intake of six typical Mediterranean foods (pasta, legumes, fish, Mediterranean vegetables, fruits, and olive oil), low consumption of four non-Mediterranean foods (red meat, butter, soft drinks, and potatoes), and limited alcohol intake. One point was given if the consumption of Mediterranean foods fell within the highest tertile, while all other intakes were assigned zero points. Similarly, one point was awarded if the intake of non-Mediterranean foods was in the lowest tertile. Alcohol intake was assigned one point if it was up to 12 g per day; both nondrinkers and individuals who consumed more than 12 g per day received zero points [24]. Patients were categorized into three groups based on the tertiles of their IMI scores as follows: Tertile I (score 0–3), low MD adherence; Tertile II (score 4–5), moderate MD adherence; Tertile III (score 6–11), high MD adherence.

### 2.2. Laboratory Measurements

Blood samples were drawn from a brachial vein after an overnight fast between 8 a.m. and 9 a.m. and directly processed. Ordinary laboratory tests included a complete blood count and assessments of serum glucose, total cholesterol, triglycerides, protein electrophoresis, creatinine, blood urea nitrogen, erythrocyte sedimentation rate (ESR), C-reactive protein (CRP), fibrinogen, and ferritin.

Levels of oxidized (GSSG) and reduced (GSH) glutathione in whole blood were measured as described previously [25]. Plasma fluorescent adducts originated from peroxidation-derived aldehydes (HNE and MDA), and proteins were quantified using spectrofluorimetry, according to established protocols [26].

Serum levels of cytokines and growth factors, including IL-1α, IL-1β, IL-2, IL-4, IL-6, IL-8, IL-10, interferon (IFN)-γ, tumor necrosis factor (TNF), vascular endothelial growth factor (VEGF), and epidermal growth factor (EGF), were assessed using an EV 3513 cytokine biochip array and competitive chemiluminescence immunoassays (Randox Laboratories Ltd., Crumlin, UK) according to the manufacturer’s instructions, with the Randox Evidence Investigator [27].

### 2.3. RNA Isolation and Quantitative Real-Time Reverse Transcription Polymerase Chain Reaction (qRT-PCR)

Peripheral blood mononuclear cells (PBMCs) were isolated immediately after blood collection via rapid Ficoll-Histopaque centrifugation for 30 min at 900× *g*. RNA was isolated from PBMC samples using a RNeasy Kit (Qiagen, Hilden, Germany) following the manufacturer’s instructions. To minimize the in vitro impact on cell activation status, a modified gradient separation technique was employed. The RNA extracted using the RNeasy kit was stored immediately at −80 °C. RNA concentration was measured using absorption spectrophotometry, and integrity was confirmed through nondenaturing agarose gel electrophoresis. A random hexamer primer and a SuperScript III Reverse Transcriptase kit (Invitrogen, Frederick, MD, USA) were used to synthesize complementary DNA. A PCR master mix was prepared containing specific primers for superoxide dismutase 1 (SOD1) (forward, TGTGGGGAAGCATTAAAGG; reverse, CCGTGTTTTCTGGATAGAGG); catalase (CAT) (forward, GCCATTGCCACAGGAAAGTA; reverse, CCAACTGGGATGAGAGGGTA); glutathione reductase (GR) (forward, GGAGACCTCACCCTGTACC; reverse, GTCATTCACCATGTCCACC); glutathione synthetase (GS) (forward, ACCTCCACCGTATATTTGAG; reverse, TTGCCCCAG ACAGCCATCTT); and glyceraldehyde-3-phosphate dehydrogenase (GAPDH) (forward, CAAGGCTGAAACGGGAA; reverse, GCATCGCCCCACTTGATTTT). AmpliTaq Gold DNA polymerase (Applied Biosystems, Foster City, CA, USA) was added to the mixture. Real-time quantification of mRNA was performed using a SYBR Green I assay and evaluated with an iCycler detection system (Bio-Rad Laboratories, Hercules, CA, USA). The threshold cycle (CT) was determined, and relative gene expression was calculated using the following formula: fold change = 2^−Δ(ΔCT)^, where ΔCT = CT_target − CT_housekeeping and Δ(ΔCT) = ΔCT_treated − ΔCT_control.

### 2.4. Statistical Analysis

Qualitative data are presented as counts and percentages, whereas quantitative data are shown as the mean ± standard deviation (SD). The Kolmogorov–Smirnov test was used to determine the Gaussian distribution of the samples.

Differences between three groups (low vs. moderate vs. high MD adherence) were analyzed using one-way analysis of variance (ANOVA), with the Tukey–Kramer applied as a post hoc analysis for continuous variables. For categorical variables, Pearson’s Chi-squared test or Fisher’s exact test was utilized. Non-parametric tests, including the Kruskal–Wallis test followed by Dunn’s method for multiple group comparisons, were applied to non-normally distributed data.

Linear regression models were used to examine the association between circulating redox balance markers and inflammation/immune response markers. Pearson’s correlation was used for normally distributed data, while Spearman’s correlation was used for non-normally distributed data to determine the strength of linear relationships between parameters. 

A sequence of unadjusted linear regression models was implemented to explore the relationships between inflammation/immune response markers and GSSG/GSH, HNE– or MDA–protein adducts as redox balance markers. The quantitative analysis was used to assess the contribution of each biochemical parameter to changes in redox markers, based on the coefficients from multiple linear regression. 

All statistical tests were two-sided, and significance was set at *p* < 0.05. Analyses were conducted using SPSS version 23 (SPSS Inc., Chicago, IL, USA) and GraphPad Prism 9 for Windows (GraphPad Software Inc., San Diego, CA, USA).

## 3. Results

### 3.1. Baseline Characteristics

During the one-year observation period, a total of 769 hospitalized patients were considered for the study. Of these, 463 were excluded, leaving 306 patients enrolled. These patients were categorized based on their adherence to the Mediterranean diet (MD) according to the Italian Mediterranean Index (IMI) (Appendix A). Specifically, 76 patients (24.8%) were included in the low adherence group (score 0–3, I tertile), 135 patients (44.1%) in the moderate adherence group (score 4–5, II tertile), and 95 patients (31.1%) in the high adherence group (score 6–11, III tertile). Age, sex, comorbidities, polypharmacy, and most biochemical variables were similar between the groups. However, serum hemoglobin was higher in patients with high adherence to the MD compared to those with low and moderate adherence (Table 1).

### 3.2. MD Adherence and Circulating Markers of Redox Balance

Redox balance was evaluated by assessing the circulating glutathione balance and proteins that had undergone oxidative modification due to lipoperoxidative reactions. Table 2 provides detailed data on circulating markers of redox homeostasis.

The circulating reduced/oxidized glutathione ratio (GSSG/GSH) differed between patients grouped according to MD adherence (F_2, 305_ = 10.640, *p* < 0.001). Specifically, GSSG/GSH levels were the lowest in patients with low adherence to the MD compared to those with moderate and high adherence. While GSSG levels showed no significant differences between groups, blood concentrations of GSH were significantly different (F_2, 305_ = 26.058, *p* < 0.001), with the lowest GSH levels observed in patients with low MD adherence compared to the other two MD adherence groups.

The concentration of aldehyde–protein adducts also varied among groups with different levels of MD adherence (hydroxynonenal–protein adducts: F_2, 305_ = 21.829, *p* < 0.001; malondialdehyde–protein adducts: F_2, 305_ = 13.628, *p* < 0.001). Notably, the highest concentrations of both markers were found in patients with low MD adherence, compared to the other two MD adherence groups. These results indicate that low adherence to the MD is associated with high circulating markers of oxidative stress, primarily due to impaired GSH synthesis/recycling.

The gene expression analysis of key antioxidant enzymes like superoxide dismutase 1 (SOD1) and catalase (CAT), as well as enzymes involved in GSH synthesis/recycling, such as glutathione reductase (GR) and glutathione synthetase (GS), was conducted using real-time RT-PCR on mRNA extracted from PBMCs of the patient groups. As shown in Figure 1, no significant differences in the expression levels of SOD1, CAT, GR, and GS were observed between the groups.

### 3.3. MD Adherence and Circulating Markers of Inflammation/Immune Response

Inflammation and immune response were analyzed in the study participants through systemic markers, with the results presented in Table 3.

While C-reactive protein (CRP) and white blood cell count were similar across groups, other markers showed significant differences (erythrocyte sedimentation rate, ESR: F_2, 305_ = 46.120, *p* < 0.001; fibrinogen: F_2, 305_ = 10.033, *p* < 0.001; ferritin: F_2, 305_ = 6.642, *p* = 0.002; α2-globulins: F_2, 305_ = 32.486, *p* < 0.001; neutrophils: F_2, 305_ = 17.396, *p* < 0.001; lymphocytes: F_2, 305_ = 15.236, *p* < 0.001; and neutrophil-to-lymphocyte ratio, NLR: F_2, 305_ = 20.066, *p* < 0.001). These markers were reduced in the group with high MD adherence compared to the low and moderate MD adherence groups, except for the lymphocyte count, which was raised in the group with high MD adherence.

To further study immunity, serum levels of cytokines and growth factors were evaluated in the participants, with the results presented in Table 4.

While the circulating concentration of most cytokines and growth factors were similar across all groups, serum levels of interleukin-6 (IL-6) and tumor necrosis factor (TNF) were significantly different (IL-6: F_2, 305_ = 350.969, *p* < 0.001; TNF: F_2, 305_ = 375.312, *p* < 0.001), with the lowest values reported in the low MD adherence group compared to the moderate and high MD adherence groups. These findings suggest a circulating pro-inflammatory status of hospitalized patients with low adherence to the MD compared to those with moderate and high MD adherence.

### 3.4. Association between Circulating Markers of Redox Balance and Inflammation/Immune Response

A correlation matrix was generated to explore the potential relationship between circulating markers of redox balance and inflammation/immunity. The results are presented in Table 5.

Building on the results from the correlation matrix, a multivariate analysis was performed to evaluate the contribution of each inflammation/immunity marker to the variation in redox markers (Table 6).

Although no significant results were observed with GSSG/GSH as the dependent variable, a significant predictive power was noted for IL-6 on both serum HNE– and MDA–protein adducts. The linear regression analysis displayed a significant direct relationship between serum IL-6 as well as the HNE–protein (R^2^ = 0.130, F_1, 304_ = 45.36, *p* < 0.001) and MDA–protein adducts (R^2^ = 0.118, F_1, 304_ = 40.70, *p* < 0.001). 

We further examined the relationship between serum IL-6 and the circulating HNE/MDA–protein adducts according to MD adherence groups. The subgroup analysis demonstrated the most significant positive relationship between IL-6 and HNE/MDA–protein adducts in the low MD adherence group (Figure 2).

## 4. Discussion

To our knowledge, this is the first investigation to demonstrate that the level of adherence to the MD may be associated with varying alterations in circulating markers of redox balance and immunity in hospitalized old patients.

We have previously shown that old patients with low MD adherence are characterized by poor nutritional status and unfavorable changes of pro-inflammatory markers that could contribute to a longer duration of hospitalization [23]. This study reaffirms previously reported levels of adherence to the MD among old patients admitted in internal medicine wards. However, in our current cohort, we did not observe any age differences between the study groups. Higher adherence to the MD has been consistently related to a reduced risk of diabetes, cancer, neurodegenerative and cardiovascular diseases, as well as an overall lower mortality [28,29,30]. One leading hypothesis to define these beneficial associations is the property of the MD to improve redox balance due to its high polyphenol content, which provides antioxidant capacity [31,32,33]. The combined beneficial effects of diverse plant-based nutrients with antioxidant properties may elucidate why the global properties of the MD hold greater value than individual food components. In this context, observational investigations exploring the relationship between the MD and biomarkers of redox balance have yielded promising results. These studies reported a negative relationship between circulating levels of oxidized LDL and MDA and positive associations with biomarkers indicative of antioxidant defense mechanisms, such as SOD activity, glutathione peroxidase activity, and circulating GSSG/GSH [34,35]. Our findings indicate that old hospitalized patients with low adherence to the MD exhibit unfavorable circulating redox balance, evidenced by high blood GSSG/GSH and elevated serum concentrations of HNE– and MDA–protein adducts. Moreover, our data suggest that this potentially detrimental redox profile may result from a depletion of reducing compounds (antioxidants) that are not sufficiently replenished or recycled. The GSSG/GSH redox potential is a key electrochemical indicator recognized as a robust regulator of redox reactions that influence various biological processes [36,37]. GSH, a tripeptide made up of glycine, cysteine, and glutamic acid, is considered a powerful antioxidant. It is synthesized by γ-glutamylcysteine synthetase and glutathione synthetase (GS) [38]. To maintain the redox balance, GSH neutralizes oxidative molecules and is converted to GSSG; thus, adequate GSH levels are sustained by the enzyme glutathione reductase (GR) [39]. We explored the mechanisms underlying the changes in GSH and the GSSG/GSH ratio by assessing the expression of antioxidant genes, such as GR and GS, in PBMCs of the participants. However, the unreliable GSH synthesis/recycling observed in patients with low adherence to the MD was not supported by any changes in the expression of GS and GR. Additionally, our data indicated no alterations in the expression of SOD1 and CAT, which encode the two primary endogenous antioxidant enzymes. 

Low adherence to the MD is related to elevated concentrations of various circulating markers of inflammation in subjects of all ages [23,40]. Consistent with previous findings, our data demonstrate that old hospitalized patients with high MD adherence exhibit reduced concentrations of serum inflammatory biomarkers compared to those with low and moderate adherence. Among these markers, we observed that serum ferritin levels were also lower in patients with high MD adherence despite the increase in Hb concentration. In this context, we propose that the variability in serum ferritin levels between groups is more likely related to inflammation than to iron status. Interestingly, CRP levels were similar across all patient groups. Given that CRP levels increase within hours of the onset of an acute condition, it is plausible that the elevated CRP values in all participants were a result of the acute condition that necessitated their hospitalization. In addition, patients with low MD adherence showed the highest serum concentrations of the pro-inflammatory cytokines IL-6 and TNF. The impact of MD adherence on the release of acute phase proteins or pro-inflammatory cytokines remains a topic of debate. However, previous investigations have shown that adherence to the MD is negatively associated with circulating IL-6 levels [41]. The anti-inflammatory effects of the MD are well documented in several systematic reviews and meta-analyses [42,43]. These effects may be attributed to the combined properties of vegetables, fruits, and olive oil, as well as to specific compounds, such as ascorbic acid, β-carotene, α-tocopherol, omega-3 polyunsaturated fatty acids, and flavonoids. It is widely recognized that old age is marked by a unique low-grade, chronic, and “sterile” inflammatory condition known as inflammaging [15,44]. Inflammaging probably stems from an impairment between pro-inflammatory and anti-inflammatory mediators as an adaptive response to lifelong exposure to stressors [45]. In this regard, proper nutrition, particularly through the MD, could be an effective strategy to slow the age-related increase in inflammatory molecule production and promote adaptive anti-inflammatory responses, thereby reducing the risk of age-related inflammatory conditions [46]. Immunosenescence alters cell–cell interactions and signaling pathways involved in both innate and adaptive immunity [47]. Innate immunity primarily involves macrophages, neutrophils, dendritic cells, and natural killer cells, while adaptive immunity includes antigen-specific B and T lymphocytes [48]. Specifically, inflammaging seems to be driven by dysfunction in innate immune cells [49]. Our data show that old hospitalized patients with high MD adherence have low circulating neutrophils and NLR compared to low and moderate MD adherence. The NLR serves as a biomarker encompassing both branches of the immune response: neutrophils predominantly contribute to innate immunity, whereas lymphocytes are primarily involved in adaptive immunity [50]. Our results indicate that adherence to the MD may weaken the activation of innate immunity, particularly because IL-6 and TNF, which are key effectors secreted by innate immune cells, were lower in patients with higher MD adherence [51].

An impaired redox homeostasis is linked to the disruption of the inflammatory-immune response [52]. The interplay between reactive molecules and innate immune mediators is coordinated by various redox-dependent transcription factors, including the nuclear factor-kappa B (NF-κB) [53]. For the first time, this study identifies a relationship between the systemic markers of redox balance and circulating biomarkers of innate immunity as well as immune mediators in elderly hospitalized patients. This relationship is notably strong in patients with low MD adherence, as indicated by the association between IL-6 and HNE/MDA–protein adducts. While we were unable to establish a causal relationship between alterations in redox homeostasis and changes in the immune response, this association reinforces the hypothesis that adherence to the MD may influence the interplay between redox balance and the immune system in age-related conditions linked to hospitalization. Building on our initial study, our observations suggest that the effect of MD adherence on redox balance might be linked to its influence on the antioxidant system, thereby regulating innate immunity. It is also plausible that the MD could mitigate low-grade inflammation, which contributes to an excess of reactive species, thus disrupting the auto-toxic loop that promotes pathophysiological pathways in age-related diseases. It is possible that the outcomes of our study could have been influenced by other factors, such as polypharmacotherapy; however, the proportions of patients receiving multiple medications were comparable across the groups with varying levels of adherence to the MD. However, more detailed investigations are required to clarify the mechanisms by which the MD modulates redox balance and immunosenescence.

The strengths of this research include its prospective plan and the evaluation of MD adherence at hospital admission using the Italian Mediterranean Index (IMI), a comprehensive nutritional questionnaire [24]. However, this investigation has several limitations. Firstly, the restricted sample size might not have led to definitive conclusions. Secondly, the study was performed in a single center, which could introduce bias. Additionally, this investigation did not examine the association between MD adherence and the specific conditions that caused hospitalization. Another limitation is that the dietary assessment relied on a single questionnaire where patients reported their eating habits over the past year. While nutritional patterns tend to be more stable than individual nutrient intakes and are better indicators of long-term dietary habits, this approach may not fully capture the nuances of their eating behaviors [54]. Furthermore, we did not account for confounding factors such as family and socio-economic status, which could have partially influenced the results. It is also important to acknowledge that our findings may be partially influenced by lifestyle factors (such as physical activity, sleep patterns, moderation, snacking, and social interactions), which were not assessed in this study. Lastly, we were unable to assess the daily intake of micronutrients in patients with varying levels of adherence to the MD. However, while future multicenter investigations are required to confirm our observations, these results represent a determinant advancement to figure out the association between adherence to the Mediterranean diet, alterations in redox balance, and inflammation/innate immunity in elderly hospitalized patients. Although we could not definitively establish the initiating role of the Mediterranean diet in oxidative stress or immunosenescence, some assumptions can be made. The disruption of redox homeostasis and immune response contributes to the decline in multisystem efficiency that defines aging and age-related diseases. Additionally, many chronic conditions can weaken antioxidant defenses and modify immune function. Given that most older adults have comorbidities, redox balance often shifts towards an oxidative state, leading to a heightened pro-inflammatory immune response. Thus, it is plausible that long-term high adherence to the MD could mitigate this vicious cycle, preventing age-related conditions.

## 5. Conclusions

Our research emphasizes the role of the Mediterranean diet in the interaction between altered redox homeostasis and disrupted inflammation/innate immune response in elderly hospitalized patients. Given that changes in redox balance primarily depend on impaired antioxidant status, it is plausible that adherence to the Mediterranean diet could positively influence the immune response and inflammation, thereby mitigating the effects of age-related conditions. Future preclinical and clinical studies with robust methodological designs are needed to identify and evaluate specific effects associated with the Mediterranean diet in order to further advocate its adoption.

## Figures and Tables

**Figure 1 nutrients-16-03359-f001:**
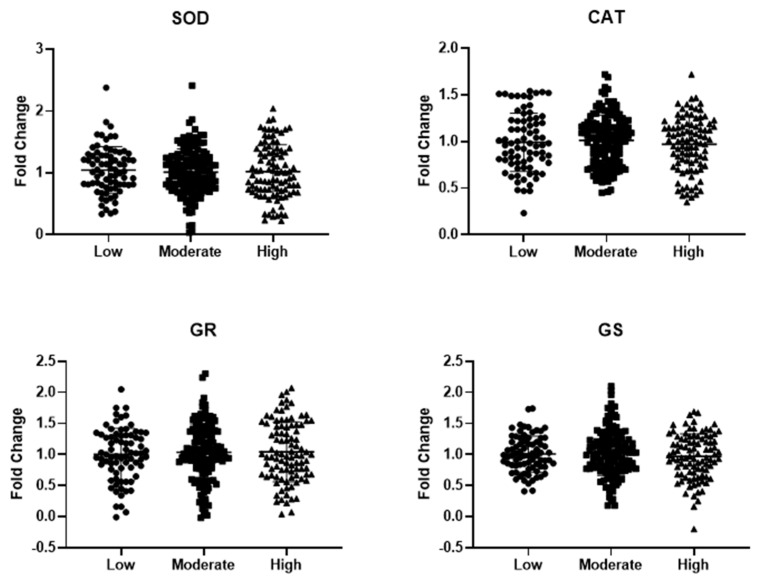
Gene expression of key antioxidant enzymes in peripheral blood mononuclear cells extracted from old hospitalized patients included in this study, grouped according to their adherence to the Mediterranean diet. Abbreviations: SOD1, superoxide dismutase 1; CAT, catalase; GR, glutathione reductase; and GS, glutathione synthetase. Data are displayed as mean ± standard deviation. Statistical analysis was performed via one-way analysis of variance.

**Figure 2 nutrients-16-03359-f002:**
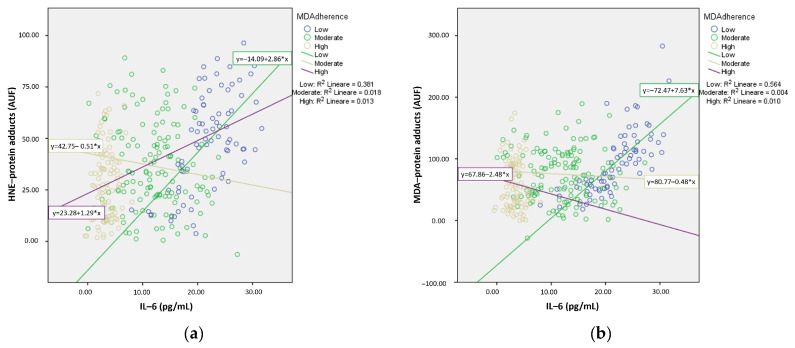
Linear regression curves showing the association between circulating levels of interleukin (IL)-6 and serum hydroxynonenal (HNE)–protein adducts (**a**) or malondialdehyde (MDA)–protein adducts (**b**). Data related to patients with different adherences to Mediterranean diet (MD adherence) were separately pooled, and the median was quantified before testing. AUF, arbitrary units of fluorescence.

**Table 1 nutrients-16-03359-t001:** Baseline characteristics of the patients included in the study, grouped in accordance with their adherence to the Mediterranean diet.

	Adherence to Mediterranean Diet			
	Low (*n* = 76)	Moderate (*n* = 135)	High (*n* = 95)	*p*
Age (years)	75.9 ± 9.82	75.4 ± 7.44	74.6 ± 6.66	0.600
Sex (M/F)	36/40	54/81	39/56	0.564
Comorbidities (*n*, %)	28 (36.8)	57 (42.2)	35 (36.8)	0.633
Polypharmacotherapy (*n*, %)	12 (15.8)	29 (21.5)	25 (26.3)	0.251
Hemoglobin (g/dL)	11.2 ± 2.03	11.7 ± 2.02	14.2 ± 3.34 ***^,†††^	**<0.001**
Glucose (mg/dL)	123 ± 47.8	119 ± 54.1	120 ± 47.8	0.819
Albumin (mg/dL)	2.99 ± 0.51	3.06 ± 0.52	3.14 ± 0.49	0.159
Total cholesterol (mg/dL)	152 ± 48.0	151 ± 49.0	146 ± 39.1	0.593
Creatinine (mg/dL)	1.19 ± 0.81	1.20 ± 0.70	1.19 ± 0.46	0.986
Blood Urea Nitrogen (mg/dL)	67.2 ± 40.7	70.2 ± 55.2	67.7 ± 40.3	0.884
Triglycerides (mg/dL)	120 ± 62.0	115 ± 49.1	128 ± 70.0	0.314

Statistical differences were evaluated by one-way analysis of variance followed by Tukey post hoc test or Pearson’s Chi-squared test. *p* values < 0.05 (in bold) were considered significant. *** = *p* < 0.001 vs. low; ^†††^ = *p* < 0.001 vs. moderate.

**Table 2 nutrients-16-03359-t002:** Circulating markers of redox balance in the patients enrolled in the study, grouped according to their adherence to the Mediterranean diet.

	Adherence to Mediterranean Diet			
	Low (*n* = 76)	Moderate (*n* = 135)	High (*n* = 95)	*p*
GSH (µM)	39.6 ± 22.1	49.0 ± 22.6 *	64.6 ± 24.7 ***^,†††^	**<0.001**
GSSG (µM)	6.39 ± 3.03	5.59 ± 2.72	5.31 ± 3.15	0.051
GSSG/GSH (%)	28.4 ± 37.6	18.2 ± 24.9 *	10.3 ± 9.23 ***	**<0.001**
HNE–protein adducts (AUF)	48.4 ± 23.1	36.6 ± 21.0 ***	27.6 ± 17.3 ***^,††^	**<0.001**
MDA–protein adducts (AUF)	93.3 ± 51.8	75.0 ± 41.6 **	59.5 ± 38.0 ***^,†^	**<0.001**

Statistical differences were evaluated by one-way analysis of variance and Tukey’s post hoc test. *p* values < 0.05 (in bold) were considered significant. * = *p* < 0.05 vs. low; ** = *p* < 0.01 vs. low; *** = *p* < 0.001 vs. low; ^†^ = *p* < 0.05 vs. moderate; ^††^ = *p* < 0.01 vs. moderate; ^†††^ = *p* < 0.001 vs. moderate. GSH, oxidized glutathione; GSSG, reduced glutathione; HNE, hydroxynonenal; and MDA, malondialdehyde.

**Table 3 nutrients-16-03359-t003:** Circulating markers of inflammation and immune response in the patients enrolled in this study grouped according to their adherence to the Mediterranean diet.

	Adherence to Mediterranean Diet			
	Low (*n* = 76)	Moderate (*n* = 135)	High (*n* = 95)	*p*
ESR (mm/h)	59.1 ± 36.7	42.7 ± 30.6 ***	18.0 ± 15.9 ***^,†††^	**<0.001**
CRP (mg/L)	49.7 ± 58.7	55.3 ± 79.2	57.8 ± 35.7	0.698
Fibrinogen (mg/dL)	408 ± 189	410 ± 167	326 ± 73.3 **^,†††^	**<0.001**
Ferritin (ng/mL)	358 ± 733	273 ± 443	103 ± 111 **^,†^	**0.002**
α2-globulins (g/dL)	12.5 ± 3.72	12.8 ± 3.67	9.42 ± 2.13 ***^,†††^	**<0.001**
White blood cells (n/mm^2^)	8945 ± 4616	8284 ± 4449	7735 ± 3553	0.180
Neutrophils (n/mm^2^)	7983 ± 1047	6125 ± 4306	3872 ± 1185 ***^,†^	**<0.001**
Lymphocytes (n/mm^2^)	1278 ± 843	1516 ± 938	1970 ± 692 ***^,†††^	**<0.001**
NLR	7.53 ± 5.88	5.89 ± 7.69	2.15 ± 0.92 ***^,†††^	**<0.001**

Statistical differences were evaluated by one-way analysis of variance and Tukey’s post hoc test. *p* values < 0.05 (in bold) were considered significant. ** = *p* < 0.01 vs. low; *** = *p* < 0.001 vs. low; ^†^ = *p* < 0.05 vs. moderate; ^†††^ = *p* < 0.001 vs. moderate. ESR, erythrocyte sedimentation rate; CRP, C-reactive protein; and NLR, neutrophil-to-lymphocyte ratio.

**Table 4 nutrients-16-03359-t004:** Circulating cytokines and growth factors in the patients enrolled in this study, grouped according to their adherence to the Mediterranean diet.

	Adherence to Mediterranean Diet			
	Low (*n* = 76)	Moderate (*n* = 135)	High (*n* = 95)	*p*
IL-1α (pg/mL)	0.53 ± 0.06	0.53 ± 0.05	0.53 ± 0.09	0.839
IL-1β (pg/mL)	0.96 ± 0.23	0.91 ± 0.29	0.92 ± 0.24	0.446
IL-2 (pg/mL)	0.99 ± 0.53	0.93 ± 0.43	0.94 ± 0.60	0.725
IL-4 (pg/mL)	0.81 ± 0.22	0.84 ± 0.16	0.84 ± 0.30	0.595
IL-6 (pg/mL)	21.8 ± 4.98	11.9 ± 5.55 ***	3.37 ± 1.56 ***^,†††^	**<0.001**
IL-8 (pg/mL)	100 ± 29.9	103 ± 35.7	99.0 ± 43.9	0.608
IL-10 (pg/mL)	1.98 ± 0.97	1.95 ± 1.17	1.69 ± 0.71	0.090
TNF (pg/mL)	13.2 ± 2.10	6.52 ± 3.36 ***	4.60 ± 1.86 ***^,†††^	**<0.001**
IFN-γ (pg/mL)	2.26 ± 0.86	2.27 ± 0.93	2.41 ± 1.12	0.485
VEGF (pg/mL)	253 ± 43.9	262 ± 35.7	261 ± 54.2	0.357
EGF (pg/mL)	13.9 ± 1.33	13.4 ± 2.04	13.6 ± 3.46	0.295

Statistical differences were evaluated by one-way analysis of variance and Tukey’s post hoc test. *p* values < 0.05 (in bold) were considered statistically significant. *** = *p* < 0.001 vs. low; ^†††^ = *p* < 0.001 vs. moderate. IL, interleukin; TNF, tumor necrosis factor; IFN, interferon; VEGF, vascular endothelial growth factor; and EGF, epidermal growth factor.

**Table 5 nutrients-16-03359-t005:** Pearson’s correlation matrix of circulating markers of inflammation, immune response, and redox balance in all the patients enrolled in the study.

	ESR	Fibrinogen	Ferritin	α2-glob	Neutro	Lympho	NLR	IL-6	TNF	GSSG/GSH	HNE	MDA
ESR	1	0.400 ***	0.469 ***	0.491 ***	0.533 ***	−0.354 ***	0.499 ***	0.556 ***	0.142 *	0.317 ***	0.310 ***	0.266 ***
Fibrinogen	0.400 ***	1	0.297 ***	0.407 ***	0.185 **	−0.18	0.120 *	0.175 **	0.213 ***	0.134 *	−0.002	0.075
Ferritin	0.469 ***	0.297 ***	1	0.166 **	0.358 ***	−0.247 ***	0.465 ***	0.250 ***	0.002	0.263 ***	0.186 **	0.069
α2-glob	0.491 ***	0.407 ***	0.166 **	1	0.287 ***	−0.176 **	0.193 **	0.318 ***	0.303 ***	0.205 ***	0.131 *	0.159 **
Neutro	0.533 ***	0.185 **	0.358 ***	0.287 ***	1	−0.242 ***	0.832 ***	0.413 ***	0.060	0.341 ***	0.260 ***	0.247 ***
Lympho	−0.354 ***	−0.18	−0.247 ***	−0.176 **	−0.242 ***	1	0.478 ***	0.370 ***	0.055	−0.208 ***	−0.282 ***	−0.185 ***
NLR	0.499 ***	0.120 *	0.465 ***	0.193 **	0.832 ***	0.478 ***	1	0.444 ***	0.019	0.481 ***	0.265 ***	0.301 ***
IL-6	0.556 ***	0.175 **	0.250 ***	0.318 ***	0.413 ***	0.370 ***	0.444 ***	1	0.381 ***	0.365 ***	0.360 ***	0.344 ***
TNF	−0.142 *	0.213 ***	0.002	−0.303 ***	−0.060	0.055	−0.019	0.381 ***	1	0.058	0.094	0.013
GSSG/GSH	0.317 ***	0.134 *	0.263 ***	0.205 ***	0.341 ***	−0.208 ***	0.481 ***	0.365 ***	0.058	1	0.205 ***	0.375 ***
HNE	0.310 ***	−0.002	0.186 **	0.131 *	0.260 ***	−0.282 ***	0.265 ***	0.360 ***	0.094	0.205 ***	1	0.233 ***
MDA	0.266 ***	0.075	0.069	0.159 **	0.247 ***	−0.185 ***	0.301 ***	0.344 ***	0.013	0.375 ***	0.233 ***	1

ESR, erythrocyte sedimentation rate; α2-glob, α2-globulins; Neutro, neutrophils; Lympho, lymphocytes; NLR, neutrophil-to-lymphocyte ratio; IL-6, interleukin-6; TNF, tumor necrosis factor; GSSG/GSH, oxidized/reduced glutathione; HNE, hydroxynonenal–protein adducts; MDA, malondialdehyde–protein adducts. * = *p* < 0.05; ** = *p* < 0.01; *** = *p* < 0.001. Notably, positive correlations were found between the serum inflammation/immunity markers and blood GSSG/GSH (except TNF), serum HNE–protein adducts (except fibrinogen and TNF), and serum MDA–protein adducts (except fibrinogen, ferritin, and TNF). Additionally, a negative correlation was reported between lymphocyte count and the three markers of redox balance.

**Table 6 nutrients-16-03359-t006:** Coefficients of multiple linear regression showing the correlations between circulating markers of redox balance (dependent variables) and inflammation/immune response markers (independent variables).

	Unstandardized Coefficients		Standardized Coefficients	*t*-Value	*p*-Value
	B	Std. Error	Beta		
Dependent: GSSG/GSH					
Constant	5.020	6.923		0.725	0.469
ESR	0.075	0.055	0.119	1.373	0.171
Fibrinogen	−0.008	0.008	−0.063	0.910	0.364
Ferritin	0.007	0.003	0.176	2.438	0.051
α2-glob	0.478	0.375	0.095	1.274	0.204
Neutro	−0.001	0.001	−0.193	−1.724	0.086
Lympho	0.001	0.002	0.041	0.551	0.582
NLR	0.496	0.369	0.164	1.344	0.180
IL-6	0.311	0.187	0.129	1.660	0.098
TNF	0.198	0.298	0.048	0.667	0.506
Model verification: ANOVA, F = 3.503; *p* < 0.001					
Dependent: HNE					
Constant	44.563	7.781		5.527	**<0.001**
ESR	0.095	0.061	0.130	1.553	0.122
Fibrinogen	−0.020	0.009	−0.143	−2.134	**0.034**
Ferritin	0.005	0.003	0.106	1.511	0.132
α2-glob	−0.206	0.421	−0.035	−0.490	0.625
Neutro	0.001	0.001	0.173	1.600	0.111
Lympho	−0.005	0.002	−0.201	−2.793	**0.006**
NLR	−0.662	0.415	−0.189	−1.595	0.112
IL-6	0.461	0.211	0.164	2.188	**0.030**
TNF	−0.374	0.334	−0.078	−1.117	0.265
Model verification: ANOVA, F = 5.711; *p* < 0.001					
Dependent: MDA					
Constant	49.287	16.046		3.053	**0.003**
ESR	0.118	0.128	0.082	0.925	0.356
Fibrinogen	−0.003	0.019	−0.009	−0.130	0.896
Ferritin	−0.004	0.007	−0.041	−0.577	0.578
α2-glob	0.441	0.875	0.038	0.504	0.615
Neutro	0.000	0.001	−0.009	−0.077	0.939
Lympho	−0.002	0.004	−0.049	−0.647	0.519
NLR	−0.114	0.861	−0.016	−0.132	0.895
IL-6	1.281	0.437	0.232	2.931	**0.004**
TNF	0.808	0.694	0.086	1.165	0.295
Model verification: ANOVA, F = 2.165; *p* = 0.025					

Significant predictive power is formatted in bold. ESR, erythrocyte sedimentation rate; α2-glob, α2-globulins; Neutro, neutrophils; Lympho, lymphocytes; NLR, neutrophil-to-lymphocyte ratio; IL-6, interleukin-6; TNF, tumor necrosis factor; GSSG/GSH, oxidized/reduced glutathione; HNE, hydroxynonenal–protein adducts; MDA, malondialdehyde–protein adducts.

## Data Availability

The data presented in this study are available upon request from the corresponding author. The data are not publicly available due to ethical and privacy restrictions.

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
