# Peer review of "Adherence to Mediterranean Diet and Biomarkers of Redox Balance and Inflammation in Old Patients Hospitalized in Internal Medicine"

_nutrients, 2024, doi:10.3390/nu16193359_

Round 1

Reviewer 1 Report

Comments and Suggestions for Authors

This investigation was designed to define the relationship between adherence to the MD and circulating markers of redox balance, while also exploring possible associations with systemic inflammation.

Therefore, it will help to more clearly present the possibility of using the Mediterranean diet to improve the health of the elderly.

Overall, it is well written. However, correction and supplementation of the following matters are required.

1. There is a need to add information on micro-nutrients in the diet used in the study.

2. There is a need to unify the form of tabulation as a whole as including table line organization,

Comments on the Quality of English Language

Minor editing of English language required.  

Author Response

This investigation was designed to define the relationship between adherence to the MD and circulating markers of redox balance, while also exploring possible associations with systemic inflammation.

Therefore, it will help to more clearly present the possibility of using the Mediterranean diet to improve the health of the elderly.

Overall, it is well written. However, correction and supplementation of the following matters are required.

Comment 1. There is a need to add information on micro-nutrients in the diet used in the study.

Response 1: We thank the reviewer for his/her positive comments, and agree with him/her about the need of information about dietary micronutrients. Nevertheless, our study was designed to assess adherence to the Mediterranean diet through the Italian Mediterranean Index. Since we did not administer food frequency questionnaires to our patients, we could not estimate daily consumption of micronutrients. This limitation is now listed in the discussion of the new manuscript (lines 410-412).

Comment 2. There is a need to unify the form of tabulation as a whole as including table line organization

Response 2: table 5 and table 6 were formatted following the reviewer’ suggestion.

Reviewer 2 Report

Comments and Suggestions for Authors

This study examined the association between adherence to the Mediterranean Diet (MD) [a list of Mediterranean foods (n=6), non-Mediterranean foods (n=4), and alcohol] and markers of redox balance and inflammation in hospitalized older adults.  The results suggest that low adherence to the MD was linked to unfavorable profiles of oxidative stress and inflammation.  These results are inline with earlier reports examining health benefits of the MD.  However, the strict focus on foods is overlooking lifestyle patterns as variables that may influence oxidative stress and inflammation (e.g., dietary habits such as moderation, snacking, whole foods vs. refined foods, or physical activity, sleep patterns and social interactions.  Hence, the final conclusion that MD is an effective ‘nutritional strategy’ for disease prevention may be misguided.  Dietary intake reflects lifestyle habits and these two practices should not be separated.  It is possible that lifestyle contributed to the favorable outcomes in this study.  This is a major study limitation that should be recognized and discussed. 

Other comments that need to be addressed:

Dietary supplements can influence oxidative stress and inflammation.  Was supplement intake information collected and controlled for in the analyses?

Polypharmacotherapy was observed in 26% of the high adherers (compared to 16% in the low adherers) – did this characteristic influence the outcomes?  Was this examined in the analyses?  A short discussion of these medications would be helpful to the reader.

It seems contradictory that Hb is elevated in the high adherers but they have a low ferritin.  Can the authors comment on this discrepancy? 

CRP values are quite high for all participants but ESR is quite low (normal) in the high adherers.  Can the authors comment on this discrepancy?

TNF is significantly elevated in the high adherers (Table 4) but in the text it is stated it is reduced (line 243 and 326).  This error must be corrected and considered in the discussion as it is counter to the current argument. 

Lines 239-245:  this text needs revising as the TNF is not following the argument. 

Table 5:  it is hard to believe that IL-6 and TNF are positively correlated based on the information in Table 4. 

Line 269:  typo ‘with’

Comments on the Quality of English Language

One typo found - otherwise English language is excellent.

Author Response

Comment 1. This study examined the association between adherence to the Mediterranean Diet (MD) [a list of Mediterranean foods (n=6), non-Mediterranean foods (n=4), and alcohol] and markers of redox balance and inflammation in hospitalized older adults.  The results suggest that low adherence to the MD was linked to unfavorable profiles of oxidative stress and inflammation.  These results are in line with earlier reports examining health benefits of the MD.  However, the strict focus on foods is overlooking lifestyle patterns as variables that may influence oxidative stress and inflammation (e.g., dietary habits such as moderation, snacking, whole foods vs. refined foods, or physical activity, sleep patterns and social interactions.  Hence, the final conclusion that MD is an effective ‘nutritional strategy’ for disease prevention may be misguided.  Dietary intake reflects lifestyle habits and these two practices should not be separated.  It is possible that lifestyle contributed to the favorable outcomes in this study.  This is a major study limitation that should be recognized and discussed. 

Response 1: we thank the reviewer for his valuable comments. This study limitation was added in the discussion section (lines 408-410).

Comment 2. Other comments that need to be addressed:

Dietary supplements can influence oxidative stress and inflammation.  Was supplement intake information collected and controlled for in the analyses?

Response 2: Our study was designed to assess adherence to the Mediterranean diet through the Italian Mediterranean Index. Since we did not administer food frequency questionnaires to our patients, we could not estimate daily consumption of micronutrients. This limitation is now listed in the discussion of the new manuscript (lines 410-412).

Comment 3: Polypharmacotherapy was observed in 26% of the high adherers (compared to 16% in the low adherers) – did this characteristic influence the outcomes?  Was this examined in the analyses?  A short discussion of these medications would be helpful to the reader.

Response 3: we agree with the reviewer’s observation about the higher proportion of polypharmacotherapy reported in patients with high adherence to the MD. Nevertheless, since statistics showed that polypharmacotherapy proportions were equal in all groups, we did not examine this parameter to check whether it influenced the outcomes. It is worth to note that patients treated with drugs potentially affecting redox status were excluded from the analysis, as reported in the Methods section (lines 78-79). To comply with the reviewer’ suggestion, we added a short discussion on this point (lines 389-392).

Comment 4: It seems contradictory that Hb is elevated in the high adherers but they have a low ferritin.  Can the authors comment on this discrepancy? 

Response 4: We thank the reviewer for this remark. Serum ferritin is mostly used as indicator of iron status, but its levels can be significantly increased in response to inflammation and/or a variety of diseases (PMID: 24549403). We hypothesize that ferritin levels in patients with high adherence to the MD were lower because the other serum inflammatory biomarkers were lesser with respect to patients with low MD adherence. This comment is now added in the discussion section (lines 338-341).

Comment 5: CRP values are quite high for all participants but ESR is quite low (normal) in the high adherers.  Can the authors comment on this discrepancy?

Response 5: Even though both CRP and ESR are used to detect inflammatory conditions, it has been shown that, when conducted simultaneously in hospital practice, these markers yield concordant results 67% of the time (PMID: 20800157). Indeed, CRP is a direct measure of inflammatory response, while ESR is an indirect measure of inflammation. CRP rises within hours of onset of an infection or inflammatory condition and returns to normal within three to seven days if the acute process is resolved. ESR, on the other hand, increases in a slower manner and remains elevated for a longer period of time. Since in our study both markers were measured at hospital admission, it is conceivable that the high values of CRP for all participants were caused by the acute condition that led to hospitalization. We added this comment in the discussion section (lines 341-344).

Comment 6: TNF is significantly elevated in the high adherers (Table 4) but in the text it is stated it is reduced (line 243 and 326).  This error must be corrected and considered in the discussion as it is counter to the current argument.

Lines 239-245:  this text needs revising as the TNF is not following the argument. 

Table 5:  it is hard to believe that IL-6 and TNF are positively correlated based on the information in Table 4. 

Response 6: We appreciate the reviewer's comments. We acknowledge that there was an error in Table 4, where the TNF values were mistakenly reversed. This mistake has now been corrected.  

Comment 7: Line 269:  typo ‘with’

Response 7: the typo is now corrected.

Round 2

Reviewer 2 Report

Comments and Suggestions for Authors

The authors have satisfactorily addressed my concerns.  Good luck with the paper.